# CMV seropositivity is a potential novel risk factor for severe COVID-19 in non-geriatric patients

Simone Weber[1][⦿], Victoria Kehl[2][⦿], Johanna Erber[3,4], Karolin I. Wagner[1], Ana-Marija Jetzlsperger[5], Teresa Burrell[1], Kilian Schober[6], Philipp Schommers[7,8], Max Augustin[7,8], Claudia S. Crowell[1,4], Markus Gerhard[1,4], Christof Winter[9], Andreas Moosmann[4,10], Christoph D. Spinner[3,4], Ulrike Protzer[5], Dieter Hoffmann[5], Elvira D'Ippolito[1‡], Dirk H. Busch[1,4‡]*

1 Institute for Medical Microbiology, Immunology and Hygiene, Technical University of Munich, Munich, Germany, 2 Institute for AI and Informatics in Medicine, School of Medicine, Technical University of Munich, Munich, Germany, 3 Department of Internal Medicine II, University Hospital Rechts Der Isar, School of Medicine, Technical University of Munich, Munich, Germany, 4 German Center for Infection Research (DZIF), Munich, Germany, 5 Institute of Virology, School of Medicine, Technical University of Munich, Munich, Germany, 6 Mikrobiologisches Institut–Klinische Mikrobiologie, Immunologie und Hygiene, Universitätsklinikum Erlangen, Friedrich-Alexander-Universität (FAU) Erlangen-Nürnberg, Erlangen, Germany, 7 Medical Faculty and University Hospital Cologne, Department I of Internal Medicine, University of Cologne, Cologne, Germany, 8 German Center for Infection Research (DZIF), Bonn-Cologne, Germany, 9 Institute of Clinicl Chemistry and Pathobiochemistry, School of Medicine, Technical University of Munich, Munich, Germany, 10 Department of Medicine III, University Hospital, LMU Munich, Munich, Germany

⦿ These authors contributed equally to this work.
‡ These authors also contributed equally to this work.
* dirk.busch@tum.de

**Data Availability Statement:** All relevant data are part of the paper.

**Funding:** This study was supported by the EIT Health CoViproteHCt #20877 and the German

## Abstract

### Background

COVID-19 has so far affected more than 250 million individuals worldwide, causing more than 5 million deaths. Several risk factors for severe disease have been identified, most of which coincide with advanced age. In younger individuals, severe COVID-19 often occurs in the absence of obvious comorbidities. Guided by the finding of cytomegalovirus (CMV)-specific T cells with some cross-reactivity to SARS-CoV-2 in a COVID-19 intensive care unit (ICU) patient, we decided to investigate whether CMV seropositivity is associated with severe or critical COVID-19.

Herpes simplex virus (HSV) serostatus was investigated as control.

### Methods

National German COVID-19 bio-sample and data banks were used to retrospectively analyze the CMV and HSV serostatus of patients who experienced mild (n = 101), moderate (n = 130) or severe to critical (n = 80) disease by IgG serology. We then investigated the relationship between disease severity and herpesvirus serostatus via statistical models.

### Results

Non-geriatric patients (< 60 years) with severe COVID-19 were found to have a very high prevalence of CMV-seropositivity, while CMV status distribution in individuals with mild disease was similar to the prevalence in the German population; interestingly, this was not

National Network of University Medicine of the Federal Ministry of Education and Research (BMBF; NaFoUniMedCovid19, 01KX2021; COVIM). E.D. was funded by the Corona-Forschungsanträge (Fakultät f. Medizin). A.M. was supported by Wilhelm Sander-Stiftung (project 2018.135.1).

**Competing interests:** D.H.B. is co-founder of STAGE Cell Therapeutics GmbH (now Juno Therapeutics/ Celgene) and T Cell Factory B.V. (now Kite/Gilead). D.H.B. has a consulting contract with and receives sponsored research support from Juno Therapeutics, a Bristol Myers Squibb Company. CDS reports grants, personal fees from AstraZeneca, personal fees and non-financial support from BBraun Melsungen, personal fees from BioNtech, grants, personal fees and non-financial support from Gilead Sciences, grants and personal fees from Janssen-Cilag, personal fees from Eli Lilly, personal fees from Formycon, personal fees from Pfizer, personal fees from Roche, other from Apeiron, grants and personal fees from MSD, grants from Cepheid, personal fees from GSK, personal fees from Molecular partners, other from Eli Lilly, personal fees from SOBI during the conduct of the study; personal fees from AbbVie, personal fees from MSD, personal fees from Synairgen, grants and personal fees from ViiV Healthcare, outside the submitted work. C.W. receives personal fees from Daiichi Sankyo and Bristol Myers Squibb. A.M. receives research support from Biosyngen.

detectable in older patients. Prediction models support the hypothesis that the CMV serostatus, unlike HSV, might be a strong biomarker in identifying younger individuals with a higher risk of developing severe COVID-19, in particular in absence of other co-morbidities.

## Conclusions

We identified 'CMV-seropositivity' as a potential novel risk factor for severe COVID-19 in non-geriatric individuals in the studied cohorts. More mechanistic analyses as well as confirmation of similar findings in cohorts representing the currently most relevant SARS-CoV-2 variants should be performed shortly.

## Introduction

Despite world-wide vaccination efforts, another wave of severe acute respiratory syndrome coronavirus 2 (SARS-CoV-2) infections is currently rapidly emerging in many countries in the northern hemisphere, bringing hospital capacities to their limits.

In the meantime, it has been well documented that individuals of advanced age and/or with certain risk factors, such as cardiovascular or pulmonary diseases, obesity as well as male sex, have a higher mortality rate in the context of SARS-CoV-2 infection [1–3]. Although multiple risk factors for severe COVID-19 disease have been identified, there seems to be a broad spectrum of disease penetrance; in addition, younger individuals with severe disease sometimes do not show any of the known risk factors. As such, the reasons for the development of severe symptoms and subsequent need for intensive care unit (ICU) admission in many patients remain unclear.

In a prior study, we investigated the phenotype of SARS-CoV-2-specific T cells in severe COVID-19 patients who required invasive mechanical ventilation, and identified T cell receptors (TCRs) specifically recognizing and reacting to the spike protein of the virus [4]. Re-expression of the identified TCRs in primary human T cells [5] allowed us to characterize the antigen reactivity profile of these SARS-CoV-2 reactive T cells in more detail. To our surprise, in follow up experiments, we could identify a strong and robust cytokine response to human Cytomegalovirus (CMV) pp65 peptide mix in different TCRs specific to SARS-CoV-2 S-protein derived from an ICU COVID-19 patient (S1A and S1B Fig). This was not the case for other herpesviruses like Epstein-Barr virus (EBV) [4]. To corroborate the CMV cross-reactivity, we further identified the exact epitope from the CMV pp65 antigen and confirmed the TCRs sensitivity by peptide titration assays (S1C–S1E Fig).

CMV is a herpesvirus that causes latently persisting infection and is transmitted through body fluids such as breastmilk or saliva. The prevalence varies geographically and is also associated with socioeconomic status [6,7]–the prevalence in Low-to-Middle-Income-Countries is generally higher than in High-Income countries. CMV seropositivity is furthermore associated with cardiovascular comorbidities as well as a higher incidence of thromboembolic events [8–11], which have already been linked to an increased risk for severe COVID-19 or have been shown to be a complication of SARS-CoV-2 infection [12]. While primary and latent CMV infections in immunocompetent individuals do not cause major symptoms, except for congenital infections in neonates in case of infection of naïve mothers, CMV (re-)activation is a feared complication in immunocompromised patients and new-borns [13–15]. Recently, a few cases of CMV reactivation in the setting of severe COVID-19 have been reported [16–19]. Intriguingly, also reactivation of EBV and Herpes simplex virus (HSV) have been described [20–23],

indicating that these latent herpesvirus infections may further contribute to the development of severe COVID-19. However, it remains to elucidate whether herpesvirus reactivations are a direct consequence of SARS-CoV-2 infections or of the treatments related to COVID-19 (for example steroids), and whether they affect the same category of patients.

Unique feature of CMV infection, differently from the other herpesviruses, is the ability to reshape the immune repertoire by creating an inflationary memory T cell response that can occupy a large fraction of the overall T cell pool [24,25], creating so-called 'memory inflation' [26]. This phenomenon becomes more prominent with increasing age, and CMV seropositivity has been linked to impaired immune responses to other infections as well as to vaccination especially in older individuals [15,27–29] presumably by immunosenescence, even if this was postulated but not demonstrated in human so far. Therefore, it was speculated that the development of effective T cell responses upon infection with SARS-CoV-2 could be strongly dampened by the presumed CMV-driven immunosenescence [30,31], which might at least in part explain the high prevalence of severe disease in the elderly (> 80 years).

Overall, the identification of SARS-CoV-2/CMV cross-reactive T cells, the known impact of CMV infection on the immune system, as well as the first reports on CMV reactivation during severe COVID-19 guided us to investigate whether CMV seropositivity is associated with severe COVID-19. In parallel, we also evaluated HSV serostatus, as HSV immunoglobulin (IgG) is prevalent in 50 to 70% of adult populations and thus should allow to recognize statistically significant effects easier than in high prevalence herpesviruses, e.g. EBV and VZV.

To address this question, CMV and HSV serostatus was retrospectively analysed via the measurement of IgG titers in cohorts of patients with mild to severe COVID-19 disease. To our surprise, these data show that CMV seropositivity is strongly associated with development of severe disease in individuals younger than 60 years, who often do not show co-morbidities. On the contrary, HSV serostatus seems to represent a risk factor for older patients (60–69 years). We could not identify such a pattern in elderly individuals (> 70 years).

## Results

To investigate the possible influence of an individual´s CMV and HSV status on the course of COVID-19, we analyzed serum samples from SARS-CoV-2 infected individuals who experienced different disease severity. CMV and HSV IgG titers were measured on a total of 311 individuals with either mild (not admitted to the hospital, n = 101, median age 50–59), moderate (hospitalized but no ICU admission, n = 130, median age 60–69) or severe to critical (ICU, n = 80, median age 70–79) disease. Where available, data on pre-existing comorbidities were also collected (Tables 1 and S1). As expected, patients who experienced more severe symptoms were of older age and/or more likely to suffer from comorbidities, with almost 90% of ICU patients being affected by at least one comorbidity (Table 1). In line with this observation, as well as with existing evidence, we also found age and comorbidities to be strong risk factors for severe COVID-19 (Table 2, univariate analyses). Furthermore, prevalence of these known comorbidities clearly rose with increasing age in our cohort (S2A Fig), thus supporting the relationship of these two variables in predicting COVID-19 outcome.

Most interestingly, CMV and HSV serostatus was also associated with higher COVID-19 severity. CMV- and HSV-seropositive individuals were more likely to be hospitalized or admitted to ICU (Table 1), and had an increased risk of developing severe COVID-19 (Table 2, univariate analyses). While we observed a tendency towards increasing percentages of CMV-seropositive individuals according to age, we did not find a dominance of CMV-positive over CMV-negative individuals in older (> 70 years) compared to younger (< 70 years) subjects (S2A Fig). This effect was the opposite for known comorbidities and HSV serostatus

**Table 1. Patient characteristics.**

| Characteristic | Severity of disease | | | | | | | |
|---|---|---|---|---|---|---|---|---|
| | Mild disease | | Hospitalization | | ICU | | Total | |
| | (N = 101) | | (N = 130) | | (N = 80) | | (N = 311)[1] | |
| Age group, n (%[2]) | | | | | | | | |
| 18–29 | 14 | (73.7) | 4 | (21.1) | 1 | (5.3) | 19 | (100.0) |
| 30–39 | 15 | (48.4) | 14 | (45.2) | 2 | (6.5) | 31 | (100.0) |
| 40–49 | 15 | (38.5) | 18 | (46.2) | 6 | (15.4) | 39 | (100.0) |
| 50–59 | 8 | (21.1) | 20 | (52.6) | 10 | (26.3) | 38 | (100.0) |
| 60–69 | 45 | (50.6) | 33 | (37.1) | 11 | (12.4) | 89 | (100.0) |
| 70–79 | 4 | (5.9) | 28 | (41.2) | 36 | (52.9) | 68 | (100.0) |
| 80–99 | 0 | (0.0) | 13 | (48.1) | 14 | (51.9) | 27 | (100.0) |
| Male, n (%) | 43 | (42.6) | 68 | (52.3) | 54 | (67.5) | 165 | (53.1) |
| CMV-reactive, n (%) | 44 | (43.6) | 94 | (72.3) | 62 | (77.5) | 200 | (64.3) |
| HSV-reactive, n (%) | 72 | (71.3) | 120 | (93.8) | 76 | (96.2) | 268 | (87) |
| Cardio-vascular co-morbidity, n/N (%) | 8/100 | (8.0) | 69/130 | (53.1) | 59/80 | (73.8) | 136/310 | (43.9) |
| Respiratory co-morbidity, n/N (%) | 5/100 | (5.0) | 16/130 | (12.3) | 13/80 | (16.3) | 34/310 | (11.0) |
| Nephrological co-morbidity, n/N (%) | 0/61 | (0.0) | 21/130 | (16.2) | 16/80 | (20.0) | 37/271 | (13.7) |
| Diabetes mellitus, n/N (%) | 4/100 | (4.0) | 27/130 | (20.8) | 22/80 | (27.5) | 53/310 | (17.1) |
| Any comorbidity, n/N (%) | 21/100 | (21.0) | 89/130 | (68.5) | 71/80 | (88.8) | 181/310 | (58.4) |

Percentages are calculated using the available data over each severity of disease group (column percent), unless otherwise stated.

[1] For HSV serology, serum from only 308 patients was available.

[2] Percentages for age group are calculated over the age groups (row percent).

(S2B and S2C Fig). These observations suggested CMV serostatus as a risk factor independent of age. In support of this interpretation, CMV seropositivity remained a significant predictor of unfavorable prognosis after including age ($OR_{Hosp}$ = 3.1, $OR_{ICU}$ = 5.0; both $p < 0.001$) and comorbidities ($OR_{Hosp.}$ = 3.1, $OR_{ICU}$ = 5.2; both $p < 0.001$) in the multinomial logistic regression model. Similar results were found for HSV serology (age—$OR_{Hosp}$ = 3.6, $OR_{ICU}$ = 4.5; both $p < 0.01$; comorbidities $OR_{Hosp.}$ = 4.8, $OR_{ICU}$ = 6.5; both $p < 0.05$) (Table 2, multivariate models).

Looking at CMV serostatus within different disease severities and decades of age further demonstrated that particularly younger patients who required admission to the ICU were mostly CMV seropositive, while this finding weakened with increasing age (Fig 1). Remarkably, all but one patient younger than 70 years admitted to the ICU and most hospitalized patients were CMV seropositive. On the contrary, we observed that almost all ICU patients were found HSV seropositive, regardless of the age, and that hospitalized patients showed a trend of increased HSV seroprevalence according to age, unlike CMV serostatus. Both CMV and HSV prevalence in the mild disease subgroup was similar to the age-matched healthy population in Germany [32], except for the very young individuals in regards to HSV seropositivity (Fig 1, S1 Table).

Classification tree models are known for their ability to identify and graphically display interactions between predictors in a straighter forward way than logistic regression. Important to us was the ability of those models to branch different subpopulations (younger *versus* older patients) using different predictors. Thus, we built the tree-counterpart of the multivariate multinomial logistic model 1 from Table 2 (Fig 2). Our study cohort was first split according to age and, secondly, only individuals younger than 59 years were further divided according to

**Table 2. Multinomial logistic regression with dependent variable severity of disease.**

| Covariates | Severity of disease | | | | | |
| --- | --- | --- | --- | --- | --- | --- |
| | Hospitalization | | | ICU | | |
| | OR | (95% CI) | p | OR | (95% CI) | p |
| *Univariate models:* | | | | | | |
| CMV-reactive | 3.4 | (2.0, 5.9) | < 0.001 | 4.5 | (2.3, 8.6) | < 0.001 |
| HSV-reactive | 6.0 | (2.6, 13.9) | < 0.001 | 10.2 | (3.0, 35.0) | < 0.001 |
| Age group | 1.4 | (1.2, 1.6) | < 0.001 | 2.1 | (1.7, 2.7) | < 0.001 |
| Male | 1.5 | (0.9, 2.5) | 0.433 | 2.8 | (1.6, 5.2) | 0.001 |
| Cardio-vascular co-morbidity | 13.0 | (5.8, 29.0) | < 0.001 | 32.3 | (13.4, 77.7) | < 0.001 |
| Respiratory co-morbidity | 2.7 | (1.0, 7.5) | 0.065 | 3.7 | (1.3, 10.8) | 0.018 |
| Nephrological co-morbidity * | 24.2 | (1.3, 433.4) | 0.031 | 31.5 | (1.7, 584.0) | 0.021 |
| Diabetes mellitus | 6.3 | (2.1, 18.6) | 0.001 | 9.1 | (3.0, 27.7) | < 0.001 |
| Any comorbidity | 8.2 | (4.5, 15.0) | < 0.001 | 29.7 | (12.8, 69.0) | < 0.001 |
| *Multivariate model 1:* | | | | | | |
| CMV-reactive | 3.1 | (1.7, 5.6) | < 0.001 | 5.0 | (2.4, 10.5) | < 0.001 |
| HSV-reactive | 3.6 | (1.5, 8.9) | 0.005 | 4.5 | (1.2, 17.6) | 0.029 |
| Age group | 1.3 | (1.1, 1.6) | 0.003 | 2.2 | (1.7, 2.8) | < 0.001 |
| *Multivariate model 2:* | | | | | | |
| CMV-reactive | 3.1 | (1.6, 5.9) | 0.001 | 5.2 | (2.3, 12.1) | < 0.001 |
| HSV-reactive | 4.8 | (1.8, 13.1) | 0.002 | 6.5 | (1.5, 28.2) | 0.012 |
| Age group | 1.0 | (0.8, 1.3) | 0.856 | 1.5 | (1.1, 2.0) | 0.005 |
| Any comorbidity | 8.1 | (4.0, 16.7) | < 0.001 | 22.5 | (8.4, 59.9) | < 0.001 |

The reference category is: Mild disease.

* Firth Penalized Likelihood correction in two separate binary logistic regression models due to quasi-complete separation of the data; Firth's correction is not yet implemented for multinomial regression.

CMV status. Again, the CMV-positive subgroups (Node 6 and 8) contained a high percentage of patients showing moderate (hospitalized) to critical (ICU) COVID-19 severity (Node 6: 71.1% vs Node 5: 21.6%; Node 8: 90.4% vs Node 7: 28.6%). Intriguingly, HSV seropositivity stratified only individuals with middle/advanced age (Node 9 and 10) (Fig 2). Similar patterns of stratification were observed when CMV and HSV serostatus were analyzed independently each in relation to age (S3 Fig), thus further corroborating the relevance of CMV and HSV in, respectively, younger and middle/advanced age groups.

In a second classification tree model we further analyzed the predictive value of CMV/HSV serostatus in relation not only to age but also to the available comorbidities. As expected, having a known comorbidity was a predominant indicator of poorer prognosis, as most of the ICU patients were found in this group (Fig 3, Node 2). Notably, in individuals without known co-morbidities, CMV but not HSV seropositivity served as a negative predictor of outcome, independent of age (node 3 and 4).

Overall, our data raise evidence that CMV serostatus might be a very strong and independent risk factor for severe COVID-19, particularly in younger individuals.

## Discussion

In this study, we identified 'CMV- and HSV-seropositivity' as potential novel risk factors for severe COVID-19. Notably, CMV serostatus served as a predictor in patients of younger age (< 60 years) and in patients with no comorbidities, for whom risk factors are still not known.

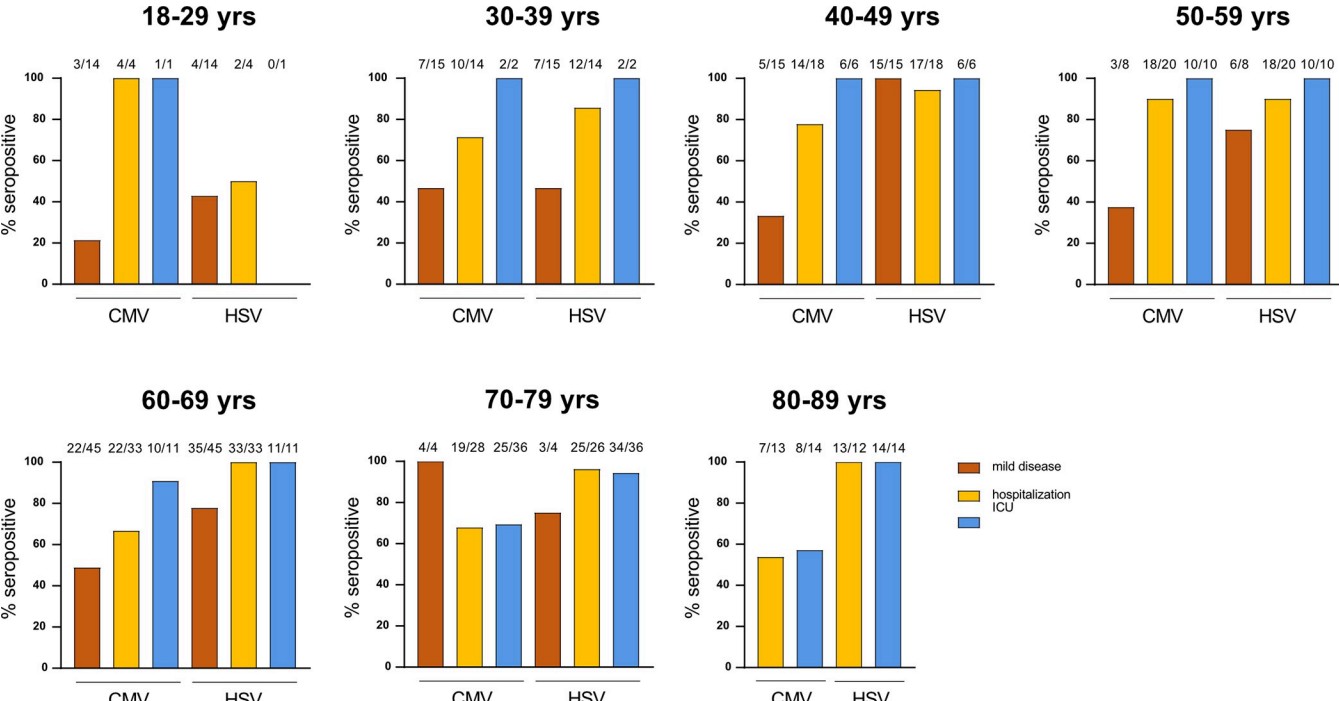

**Fig 1. CMV serology associates with severity of COVID-19 in young individuals.** CMV and HSV IgG titers were measured in serum collected from COVID-19 patients that either suffered from mild disease or required hospitalization (ICU and non-ICU, or hospitalized). Shown are percentages of CMV- and HSV-positive individuals according to age and disease severity. Numbers above bars indicate the absolute number of seropositive subjects on the total number of individuals per subgroup.

In contrast, HSV serostatus identified higher risk of severe COVID-19 in patients of middle/advanced age. Our current data cannot distinguish whether seropositivity to these two herpesviruses is just a biomarker or more directly involved in the pathophysiology of severe COVID-19. Further research in this direction should be rapidly performed, as the underlying mechanisms might also open up novel options for therapy improvement.

The identification of CMV/SARS-CoV-2 cross-reactive T cells (S1 Fig) might indicate that CMV infection could be indirectly involved in severe COVID-19 via the preferential recruitment of T cells from the antigen-experienced or memory T cell pool. Such T cells are often less reactive to the antigen for which they were not originally primed and, because of this, an impaired T cell response could fail to control SARS-CoV-2, thereby leading to severe COVID-19 [33]. Due to the phenomenon of 'memory inflation', CMV-specific T cells often dominate the general memory T cell population, especially in older CMV-seropositive individuals where the pool of naïve T cells narrows. Therefore, CMV-specific T cells might have a higher likelihood of participating in the pool of recruited SARS-CoV-2 specific T cells from cross-reactive repertoires. But this phenomenon is certainly not restricted to CMV. Cross-reactivity to SARS-CoV-2 epitopes in severe COVID-19 patients has also been shown for other target specificities, such as other common cold coronaviruses [34–40]. Many groups world-wide, including ourselves, are currently trying to shed more light on the relevance of recruitment of SARS-CoV-2-specific T cells from cross-reactive antigen-experienced T cell repertoires for severe COVID-19, and CMV might be a "master factor" in this context considering its extreme impact on T cell repertoire shifts. However, with the existing body of data postulating that CMV supports immunosenecence especially in elderly individuals, it remains surprising that our current study on COVID-19 identified a correlation between CMV seropositivity and

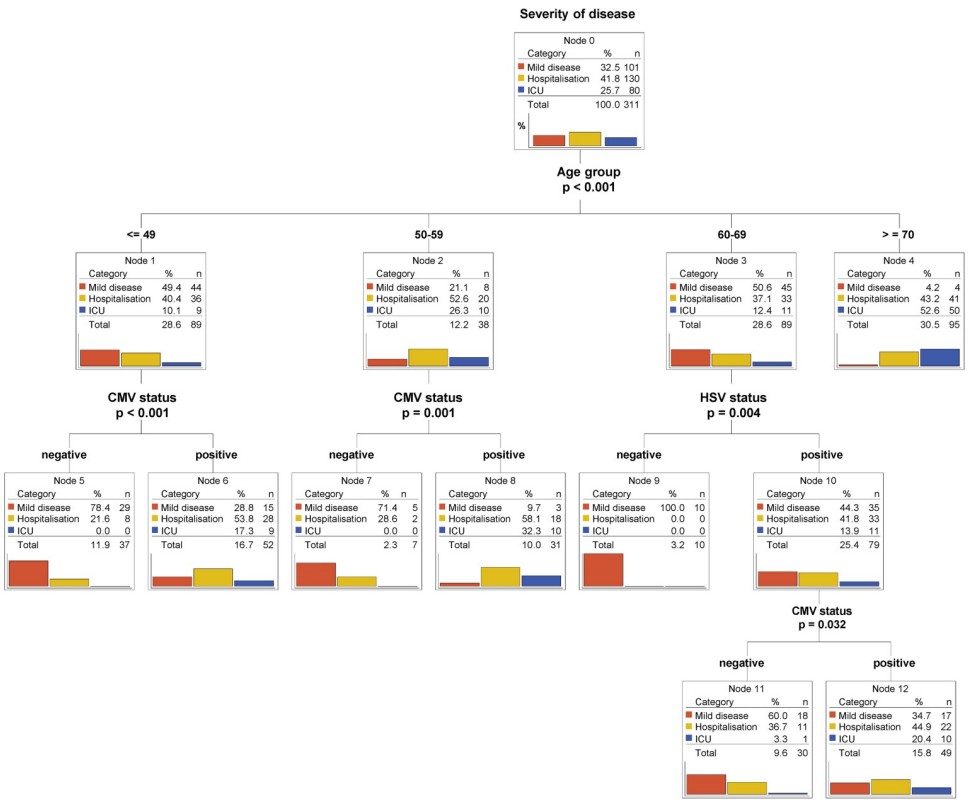

**Fig 2. CMV and HSV serostatus predicts outcome in different age groups.** Classification tree model (CHAID) using CMV serostatus, HSV serostatus and age as predictors of severity of disease. Bar plots represent percentages. Percentages for categories (mild disease, hospitalization and ICU) are calculated within the node. Percentages for the totals are calculated using the entire dataset.

disease severity particularly for younger patients. If CMV seropositivity would indeed impair the quality of SARS-CoV-2 specific T cells responses in severe COVID-19, adoptive T cell therapy with highly SARS-CoV-2-specific T cells might become an interesting option to therapeutically compensate for the defect. Indeed, first clinical trials in this direction are currently ongoing and recent trials based on adoptive transfer of memory T cells from convalescent donors have shown some promising results [41].

A completely different scenario would be a more direct involvement of CMV in severe COVID-19 pathogenesis of younger individuals via CMV reactivation. Few recent case reports have described CMV-reactivation during SARS-CoV-2 and postulated that CMV-driven pneumonitis might have been a key driver of lung function compromise and clinical outcomes in these COVID-19 patients [16,17,19]. Pathophysiologically, inflammatory cytokines stimulated by SARS-CoV-2 could lead to the reactivation of latent CMV residing in the lung. We have tried searching retrospectively in our cohort for evidence of CMV reactivation (e.g. via CMV PCR in bronchoalvelolar lavages), but so far failed to demonstrate more clear evidence for reactivation. Unfortunately, these results are not conclusive, since demonstration of CMV reactivation is complex and requires optimal sample acquisition and diagnostics. We are currently initiating prospective studies to specifically search for evidence of CMV reactivation during severe COVID-19.

HSV reactivation has been more often described in critically-ill COVID-19 patients, despite its impact on hospital mortality is still controversial [22,23,42]. Moreover, HSV reactivation

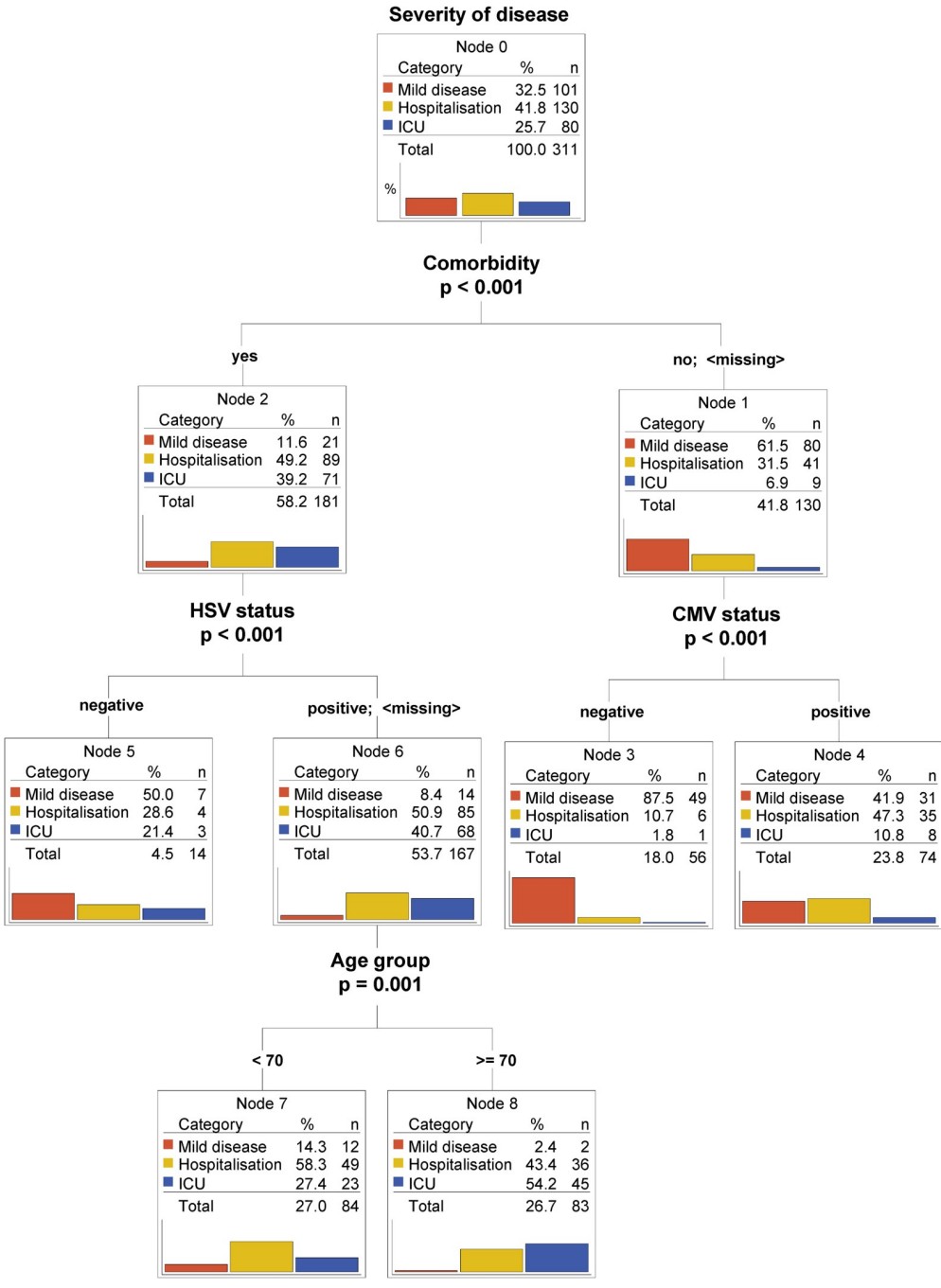

**Fig 3. CMV serostatus remains an independent predictor of worse outcome for young patients with no comorbidities.** Classification tree model (CHAID) using CMV serostatus, HSV serostatus, comorbidities and age as predictors. Bar plots represent percentages. Percentages for categories (mild disease, hospitalization and ICU) are calculated within the node. Percentages for the totals are calculated using the entire dataset.

was associated to the length of stay on ICU and mechanical ventilation [42]. However, HSV reactivation seems to broadly occur in immunocompetent patients with acute respiratory distress syndrome in relation to ICU admission [43]. Combined with the observation that HSV seroprevalence in our COVID-19 diseased population stabilizes at 90–100% already from the

age of 40 years old, it is conceivable to hypothesize that HSV reactivation might be a more general consequence of the ICU care rather than a pathophysiologically contribution to COVID-19. Still, a deeper understanding of the impact of these latent infections may help in a more tailored patient monitoring and treatment.

Although our study shows surprising results that are possibly impactful for COVID-19 patients' outcomes, there are also some limitations that should be mentioned. Our cohort comprises patients and biological samples that were collected in Germany earlier in the pandemic. Therefore, it is important to initiate similar studies with additional subjects to confirm whether our findings can be generalized to patients from other countries. Also, socioeconomical factors should be taken into consideration, considering that both CMV seroprevalence and severity of SARS-CoV-2 infections have been linked to lower socioeconomic status [44–46]. Additionally, the biomaterial was collected before the emergence of variants of concern that are currently dominating the pandemic (e.g. delta variant and omicron in Europe) and before the global vaccination campaign. Thus, it will be important to perform follow-up analyses in settings that also render the current infection and vaccination dynamics. Another limitation of our study is that the different patient subgroups are not fully balanced by age and gender–which is partly due to biological reasons (for example absence of mildly symptomatic elderly individuals > 80 years). As the biomaterial and patient data used for our analyses were collected in the context of different study protocols, availability of data varied. All of these factors added some challenges to the statistical analyses; however, despite these limitations, the main findings summarized in this report remain robust and highly significant.

In summary, we identified 'CMV-seropositivity' as a novel risk factor for severe COVID-19 in younger individuals. Our findings may have immediate implications on patient management and inspire investigation into SARS-CoV-2 vaccine response quality with respect to CMV serostatus in more detail.

## Materials and methods

### Clinical samples

For mildly symptomatic SARS-CoV-2 infections, blood samples were collected at the Helios Klinikum München West (n = 39), from healthcare employees who were diagnosed via PCR and experienced mild symptoms (cold, cough and mild fever), but did not require hospitalized treatment at any time. Additional biosamples from mildly diseased patients were acquired from the university hospital Köln in the context of the Nationales Netzwerk Universitätsmedizin consortium (n = 62). Hospitalized patients (ICU, n = 80 and non-ICU, n = 130) were prospectively included in the COVID-19 registry COMRI at the University Hospital rechts der Isar. Serum samples were collected according to the study protocol. Clinical data were retrospectively collected by medical chart review.

All participants provided informed written consent. Approval for the study design and sample collection was obtained from the local ethics committee of the Technical University of Munich (reference number 182/20 and 633/21 S-SR) and the COVIM steering committee.

### Cell isolation and culture conditions

PBMCs were isolated from whole blood by gradient density centrifugation according to manufacturer's instructions (Pancoll human) and either frozen at -80˚C in a freezing medium composed of 90% FCS and 10% DMSO. PBMCs were cultured in RPMI 1640 supplemented with 10% FCS, 0.025% l-glutamine, 0.1% HEPES, 0.001% gentamycin, 0.002% streptomycin (complete RPMI) and 180 U/ml IL-2. Jurkat-based triple parameter reporter cells (J-TPR) [47] were cultured in complete RMPI.

CD40 activated B cells were generated according to Wiesner et al. [48], and cultured in complete RPMI with addition of 2 ng/ml IL-4 (and 1 μg/ml Cyclosporin A in the early phase of cultivation). In brief, thawed autologous donor PBMCs were co-cultured with murine fibroblastic L cells stably transfected with the human CD40 ligand gene. Plates for coincubation were prepared by plating irradiated (180 Gy) CD40L-expressing L cells at $1.0 \times 10^6$ cells per 12-well or 96-well plate one to three days prior to coincubation.

All cells were cultured in a humidified incubator at 37°C and 5% $CO_2$.

## TCR DNA template design and CRISPR/Cas9-mediated TCR knock-in

DNA constructs for CRISPR/Cas-9-mediated HDR at TRAC locus were designed *in silico* with the following structure: 5′ homology arm (300–400 base pairs), P2A, TCR-β (including mTRBC with additional cysteine bridge), T2A, TCR-α (including mTRAC with additional cysteine bridge), bGHpA tail, 3′ homology arm (300–400 base pair). All HDR DNA template sequences were synthesized by Twist.

CRISPR/Cas9-mediated endogenous TCR knock-out and transgenic TCR knock-in (KI) was performed as described [5]. Briefly, freshly isolated PBMCs were activated with CD3/CD28 Expamer (Juno Therapeutics), 300 U/ml IL-2, 5 ng/ml IL-7 and 5 ng/ml IL-15. After removing of the stimulus by incubation in a Biotin solution (1 mM), cells were electroporated in a Nucleofector Solution containing Cas9 ribonucleoprotein and DNA templates with a 4D Nucleofector XL unit (Lonza). After electroporation, cells were cultured in RPMI with 180 IU/ml IL-2 before analysis. For J-TPRs, cells were seeded at a density of $0.1 \times 10^6$ cells/ml two days prior editing, and processed as described before for PBMCs. Cells were sorted on a MoFlo Astrios EQ cell sorter prior to functional assays.

## CMV epitope deconvolution

To determine the epitope specificity of the analyzed clonotypes we used an overlapping peptide bank spreading over the whole pp65 sequence with 15mer peptides overlapping in 11 amino acids [49]. Via the arrangement of the peptides in a two-dimensional matrix of subpools that each overlap in exactly one peptide, the epitope specify is identified via the reactivity to two of the subpools.

## Antigen-specific T cell stimulation assays

TCR-engineered PBMCs were stimulated with the peptide pool of interest (PepTivator® SARS-CoV-2 Prot_S from Miltenyi Biotech or PepMix™ HCMVA (pp65) from JPT) at a concentration of 1 μg/ml. For TCR-engineered T cells, autologous antigen presenting cells (PBMCs) were loaded with the different peptide mixes via incubation for 2 h at 37°C, and co-cultured with engineered T cells in a 1:1 effector:target ratio. Unpulsed PBMCs served as negative control whereas 25 ng/ml PMA and 1 μg/ml Ionomycin served as positive control. After incubation for 4 h at 37°C in presence of 1 μg/ml GolgiPlug (Brefeldin A), cells were stained with EMA solution (1:1000) for live/dead discrimination and subsequently with surface antibodies: anti-CD8-PE (1:200, eBioscience, clone OKT8), anti-CD3-BV421 (1:100, BD Biosciences, clone SK7) and anti-murine TCR β-chain-APC/Fire750 (1:50, BioLegend, clone H57-597). Cells were fixed using Cytofix/Cytoperm solution followed by staining for intracellular cytokines by anti-IFN-γ-FITC antibody (1:10, BD Pharmingen, clone 25723.11) and anti-IL-2-APC (1:25, BD Pharmingen, clone 5.344.111).

For J-TPR assays, autologous CD40 activated B cells were loaded with either one of the 24 peptide pools of the CMV pp65 antigen or with different concentrations of the pp65-derived epitope AGILARNLVPMVATV ($10^{-12}$, $10^{-11}$, $10^{-10}$, $10^{-9}$, $10^{-8}$, $10^{-7}$, $10^{-6}$, $10^{-5}$, $10^{-4}$ M) for 2

h at 37˚C. Pulsed CD40L activated cells were then co-cultured with TCR-engineered J-TPRs cells in a 1:5 effector:target ratio. Unpulsed BBLs served as negative control whereas 25 ng/ml PMA and 1 μg/ml Ionomycin served as positive control. After incubation for 18 h at 37˚C, cells were stained with the surface antibodies: anti-murine TCR β-chain-APC (BioLegend, Clone H57-597), anti-CD4-PE (Life Technologies, Clone RPA-T4) and anti-CD19-ECD (Beckman Coulter, clone J3-119), each used at a dilution of 1:100. NFAT-GFP and NFκB-CFP reporter expression was directly analyzed via flow cytometry.

Flow cytometric analysis was performed on the CytoFlex S Cell Analyzer.

## CMV and HSV serology

Analyses were conducted at the Institute for Virology, Technical University Munich. CMV IgG was measured in serum samples with a chemiluminescent microparticle immunoassay on Architect i1000 (Abbott GmbH, Wiesbaden). The cut-off value was 6 AU/ml. HSV IgG was measured with the chemiluminescent immunoassay HSV-1/2 IG on the Liaison platform (Dia-Sorin GmbH, Dietzenbach). Results $> = 1.1$ were considered positive.

## Statistical methods

Descriptive statistics are provided as absolute and relative frequencies by severity of disease and in total. Information about patient age was collected on an ordinal scale. Univariate and multivariate multinomial logistic regression models were calculated using "mild disease" as reference category of the dependent variable severity. Due to quasi-complete separation of the data, some models needed a Firth Penalized Likelihood correction. This solution is available only for the binary logistic regression, which is why the two binary logistic regressions were calculated instead of one multinomial logistic regression. The odds ratios (OR) are presented together with their 95% CI and the corresponding p-value. In addition, classification tree models (CHAID) were built from all available data using the following specifications: dependent variable severity of disease, pearson chi$^2$ statistic for the split, Bonferroni-adjusted p-values, 10-fold cross validation, and minimum number of cases in a parent node 20; in a child node 7. The significance level was set to 5%. Analysis was performed using IBM SPSS version 26 (IBM Corp., Armonk, N.Y., USA) and SAS 9.4 (SAS Institute Inc., Cary, NC, USA).

## Supporting information

**S1 Fig. Cross-reactivity of SARS-CoV-2-specific TCRs to CMV.** A-B) TCRs were isolated from an ICU patient and engineered into PBMCs from healthy donors via CRISPR/Cas9-mediated knock-in. Engineered T cells were co-cultured with autologous PBMCs previously pulsed with 1 μg/ml Peptivator S mix or CMV pp65 mix for 4 h at 37˚C. Shown are representative raw data (A) and quantification (B) of IL-2 and IFN-γ production. C) Schematic depiction of the J-TPR system. Briefly, fluorescent protein genes were engineered downstream to TCR-triggered transcription factors. T cell activation can therefore be monitored by activation of the reporter genes. D) Overlapping peptides are generated from the CMV pp65 antigen and pooled into 24 subpools. Depicted is a summary heat map showing NFAT responses of TCR-engineered J-TPR cells after 18 h of co-culture with autologous CD40 activated B cells pulsed with 1 μg/ml of each individual subpool. The epitope AGILARNLVPMVAT is the one shared among pool 3 and 24. E) TCR-engineered J-TPR cells were co-cultured with autologous CD40 activated B cells pulsed with different AGILARNLVPMVAT peptide concentrations for 18 h at 37˚C. Shown are NFAT reporter EC$_{50}$ curves (left) and quantification (right).
(TIFF)

**S2 Fig. Occurrence of CMV, HSV and comorbidities according to age.** Bar graphs showing the percentage of individuals enrolled in this study positive or negative for CMV (A) and HSV serostatus (B), and with or without comorbidities (C). Numbers within the bars indicate absolute numbers of individuals.
(TIF)

**S3 Fig. CMV and HSV serostatus predicts outcome in different age groups.** Classification tree model (CHAID) using age and either CMV serostatus or HSV serostatus as predictors of severity of disease. Bar plots represent percentages. Percentages for categories (mild disease, hospitalization and ICU) are calculated within the node. Percentages for the totals are calculated using the entire dataset.
(TIF)

**S1 Table.**
(XLSX)

## Acknowledgments

E.D., D.H.B., S.W. conceptualized the study; S.W. performed experiments and data analyses; V.K. performed and described statistical analyses; T.B., P.S., M.A., C.S.C. collected samples and clinical information of mild COVID-19; J.E., C.W, S.D.S designed and organized the study on hospitalized COVID-19 patients; A.M.J., D.H. and U.P. performed CMV serology measurements; A.M. supported epitope screenings; K.I.W., K.S., A.M.J. provided resources; M.G, D.H.B, E.D. acquired funding; E.D., V.K. prepared figures and tables; S.W., D.H.B wrote the manuscript; D.H.B. and E.D. supervised the study and administered the project. All authors read and approved the manuscript.

## Author Contributions

**Conceptualization:** Simone Weber, Elvira D'Ippolito, Dirk H. Busch.

**Formal analysis:** Simone Weber, Victoria Kehl.

**Funding acquisition:** Markus Gerhard, Dirk H. Busch.

**Investigation:** Simone Weber, Victoria Kehl, Johanna Erber, Ana-Marija Jetzlsperger, Claudia S. Crowell, Christof Winter, Christoph D. Spinner, Ulrike Protzer, Dieter Hoffmann.

**Methodology:** Victoria Kehl, Dieter Hoffmann.

**Project administration:** Elvira D'Ippolito.

**Resources:** Johanna Erber, Karolin I. Wagner, Ana-Marija Jetzlsperger, Teresa Burrell, Kilian Schober, Philipp Schommers, Max Augustin, Markus Gerhard, Christof Winter, Andreas Moosmann, Christoph D. Spinner, Ulrike Protzer.

**Supervision:** Elvira D'Ippolito, Dirk H. Busch.

**Writing – original draft:** Simone Weber, Victoria Kehl, Dirk H. Busch.

**Writing – review & editing:** Elvira D'Ippolito.

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
