## [Decision Letter · Decision Letter 0]

16 Mar 2022

PONE-D-22-02057CMV seropositivity is a potential novel risk factor for severe COVID-19 in non-geriatric patientsPLOS ONE

Dear Dr. Busch,

Thank you for submitting your manuscript to PLOS ONE. After careful consideration, we feel that it has merit but does not fully meet PLOS ONE’s publication criteria as it currently stands. Therefore, we invite you to submit a revised version of the manuscript that addresses the points raised during the review process.

Your manuscript has been evaluated by three expert reviewers whose comments are appended below.  In general they found your study interesting and worthy of publication, but there were concerns about whether the correlations with CMV were causal and whether they were in fact specific to CMV.   The major issue is whether there may also be correlations with other herpesviruses such as HSV or EBV, and it is recommended that you evaluate patient sera for IgG to these viruses and include that data.  Other issues can mainly be addressed through textual clarification.  Please address all of the reviewer's comments in your rebuttal letter with the revised manuscript.  

We look forward to receiving your revised manuscript.

Kind regards,

Juliet V Spencer, Ph.D.

Academic Editor

PLOS ONE

Journal Requirements:

[This study was supported by the EIT Health CoViproteHCt #20877 and the German National 

Network of University Medicine of the Federal Ministry of Education and Research (BMBF; 

NaFoUniMedCovid19, 01KX2021; COVIM).]

 [Yes - This study was supported by the EIT Health CoViproteHCt #20877 and the German National

Network of University Medicine of the Federal Ministry of Education and Research (BMBF;

NaFoUniMedCovid19, 01KX2021; COVIM).]

[Authors with competing interests D.H.B. is co-founder of STAGE Cell Therapeutics GmbH (now Juno Therapeutics/ Celgene) and T Cell Factory B.V. (now Kite/Gilead). D.H.B. has a consulting contract with and receives sponsored research support from Juno Therapeutics, a Bristol Myers Squibb Company. C.D.S reports grants and personal fees from AbbVie, grants, fees and non-financial support from Gilead Sciences, grants and personal fees from Janssen-Cilag, grants and personal fees from MSD, grants from Cepheid, personal fees from GSK, grants and personal fees from ViiV Healthcare, during the conduct of the study; fees from AstraZeneca, other from Apeiron, grants, personal fees and non-financial support from BBraun Melsungen, grants, personal fees from Eli Lilly, personal fees from Formycon, personal fees from Molecular partners, grants and personal fees from Eli Lilly, personal fees from SOBI. The other authors have no financial conflicts of interest.]

5. Please amend your manuscript to include your abstract after the title page.

Reviewers' comments:

Reviewer's Responses to Questions

**Comments to the Author**

1. Is the manuscript technically sound, and do the data support the conclusions?

Reviewer #1: Partly

Reviewer #2: Yes

Reviewer #3: Partly

2. Has the statistical analysis been performed appropriately and rigorously? 

Reviewer #1: Yes

Reviewer #2: Yes

Reviewer #3: I Don't Know

3. Have the authors made all data underlying the findings in their manuscript fully available?

Reviewer #1: Yes

Reviewer #2: Yes

Reviewer #3: Yes

4. Is the manuscript presented in an intelligible fashion and written in standard English?

Reviewer #1: Yes

Reviewer #2: Yes

Reviewer #3: Yes

5. Review Comments to the Author

Reviewer #1: In the here presented study by Weber et al., the authors investigate whether CMV serostatus can affect the outcome and severity of a subsequence SARS CoV-2 infection as a predictive correlation of CMV serostatus and hospitalization or severe outcome during Covid-19 disease could influence observation and treatment decisions in hospital setting worldwide. The authors do observe, that younger patients that go on to develop mild symptoms are almost exclusively CMV-seronegative whereas the CMV positive population seems to have a more severe outcome after SARS CoV-2 infection with increased numbers of hospitalizations and ICU visits. Due to increasing numbers of co-morbidities, this statistical significance does not seem to be present in the oldest age group in this study, indicating the CMV serostatus might be especially important for younger otherwise healthy patients, for which the largest effect of serostatus and disease outcome could be observed. The data presented in this study is highly interesting and unsurprisingly very relevant, but while the authors try to explain their sole focus on CMV with the unique immunology describe for this virus, which could affect the adaptive immunity in CMV positive individuals, I believe it would be important to include additional, hopefully non-significant, controls into the study as the claims made by the authors and the potential significance for the here presented data on the filed requires adequate data to strengthen the here postulated hypothesis. I think I have repeated myself in the following more detailed analysis a few times, but here are some changes I would advise:

1) “To our surprise, in follow up experiments we could identify a strong and robust cytokine response to human Cytomegalovirus (CMV) pp65 peptide mix in different TCRs specific to SARS-CoV-2 S-protein derived from an ICU COVID-19 patient (Supplementary Fig.1 a-b).”

While I have no problem believing that this data is true, what makes the authors think that this is something specific to CMV? Could there be cross reactive T-cells to other herpesviruses? HSV-1 or EBV for instance? Could they have a similar effect? Honestly, I would likely advise the authors to also test all the sera shown in figures 1, 2 and 3 for IgG against other herpesviruses for the purpose of demining if they also might correlate with severe Covid-19 disease or not. At a minimum, this would serve as a control for their CMV studies showing that what they see is not universally true for all tested pathogens.

2) “While primary and latent CMV infections in immunocompetent individuals do not cause major symptoms, CMV (re-)activation is a feared complication in immunocompromised patients and new-borns13–15”.

While CMV re-activation is clearly a major health concern in transplant recipients and can be problematic in congenital infections, primary infection of CMV naïve mothers resulting in congenital infection are probably the more impactful threat to neonates.

3) “Recently, a few cases of CMV reactivation in the setting of severe COVID-19 have been reported16–19.”

This statement is a little unclear. Have the CMV re-infections been observed as the results of SARS CoV-2 infections and the resulting Covid-19 disease, or could these reactivations have been the results of the treatment the patients received as a result of their condition which could have included steroids and hence might have cause an immunosuppressive environment in these individuals, which might have resulted in CMV re-activation as well as potentially the re-activations of other herpesviruses.

4) “Therefore, it was speculated that the development of effective T cell responses upon infection with SARS-CoV-2 could be strongly dampened by CMV-driven immunosenescence26,27, which might at least in part explain the high prevalence of severe disease in the elderly (>80 years).”

This is a very hypothetical statement, to my knowledge, CMV induced immunosenescence is a concept that has been postulated for humans, but has so far only been shown in inbred rodent model systems and is still a matter of some controversy. It is probably more likely that for the data presented here, co-morbidities and overall health might play a more significant role.

5) “Overall, the identification of SARS-CoV-2/CMV cross-reactive T cells, the known impact of CMV infection on the immune system, as well as the first reports on CMV reactivation during severe COVID-19 guided us to investigate whether CMV seropositivity is associated with severe COVID-19.”

This is in my opinion very circumstantial evidence for an involvement of CMV in determining disease severity after SARS COV-2 infection. While the data generated in this study does look interesting, I would advise including other viruses (herpesviruses) as controls to clearly demonstrate the unique role and biology of CMV.

6) “Looking at CMV serostatus within different disease severities and decades of age further demonstrates that particularly younger patients who required admission to the ICU were mostly CMV seropositive, while this finding weakened with increasing age (Fig. 1).”

While younger patients are generally in better heath than older individuals and hence have to be hospitalized less and spend less time in the ICU, does the here presented data indicate that younger individuals of lower socioeconomic status or from developing nations are at higher risk of more severe Covid-19 disease compared to their age matched peers in richer and more developed nations as they are more likely to be CMV seropositive? It’s here any data for this in the literature?

7) “Remarkably, all but one patient younger than 70 years admitted to the ICU and most hospitalized patients were CMV seropositive. Conversely, the CMV prevalence in the mild disease subgroup was similar to the age-matched healthy population in Germany28.”

While the authors show data indicating the CMV does have an effect on the severity of the disease, did it also affect the overall length of the stay in the hospital? Do the authors have any more hard virological or immunological data, e.g. viremia that would indicate that CMV status affects the subsequence Covid-19 disease progression?

8) “Intriguingly, after age stratification, younger patients suffering from comorbidities (<70, Fig 3, node 3) were more likely to develop a severe course of disease requiring ICU treatment when CMV-seropositive (CMV positive: 33.3%; CMV negative: 4.0%) (Fig 3, nodes 7 and 8). In individuals without known co-morbidities, CMV seropositivity again served as a negative predictor of outcome, but was independent of age (node 5 and 6).”

Again, these data would indicate that younger people from lower socioeconomic backgrounds, especially in the poorest nations on earth additionally exposed to other circulating diseases like Mtb and other potential co-morbidities like malnutrition that could affect the overall health and immune status, should be more prone to higher hospitalization and death rates compared to their peers in the developed world. Is there any indication that this might be true?

9) “The identification of CMV/SARS-CoV-2 cross-reactive T cells (Suppl. Fig. 1) might indicate that CMV infection is indirectly involved in severe COVID-19 via the preferential recruitment of T cells from the antigen-experienced or memory T cell pool.”

While I do believe that that could be happening, the authors do not present any data that this is specific to CMV but simply work under that assumption. As mentioned above, some controls are advised.”

10) “Such T cells are often less reactive to the antigen for which they were not originally primed and because of this an impaired T cell response could fail to control SARS-CoV-2, thereby leading to severe COVID-19.?

It would be beneficial to the reader if the authors could give a reference for this statement.

11) “Cross-reactivity to SARS-CoV-2 epitopes in severe COVID-19 patients has also been shown for other target specificities, such as other common cold corona viruses29–35.”

As mentioned before, this information should be grounds to test other herpesviruses especially EBV as an important control in this manuscript to determine if CMV is unique or not. When it comes to the T-cell responses, cross-reactive clones targeting other viruses apparently exist.

12) “Few recent case reports have described CMV-reactivation during SARS-CoV-2 and postulated that CMV-driven pneumonitis might have been a key driver of lung function compromise and clinical outcomes in these COVID-19 patients16,17,19.”

If this is true, would CMV be the result of the underlying SARS infection, or would CMV reactivation result from the steroid treatment during the severe course of infection and hence be independent of the ongoing virus infection?

Reviewer #2: The ongoing pandemic induced by infections with SARS-CoV-2 results in a wide array of disease outcomes ranging from asymptomatic to high morbidity and mortality. Multiple factors have been correlated with poor disease events including age, gender and pre-existing comorbidities. This current study seeks to characterize the impacts of seropositivity against CMV in relation to COVID disease. The authors are building on an earlier study to in which they enriched for T-cell receptors that recognize the coronavirus SPIKE protein from patients with severe COVID. The surprising finding was that there was a significant enrichment for TCR clones that and isolated clones had an increase reactivity profile for CMV antigens. This current study builds on this finding and seeks to determine the CMV seropositivity status of individuals in Germany that experience severe COVID outcomes in relation to those with milder symptoms. The authors provide compelling evidence that there is a significant correlation of CMV serostatus with poor COVID outcomes which was most evident in the younger population especially those with comorbidities. As CMV exhibits higher incidence with age, it was not surprising that a majority of the aged population with COVID were CMV seropositive. However when this group was stratified by CMV seropositive vs seronegative, there was not a significant increased risk factor for COVID hospitalization based on the presence of the herpesvirus.

It remains unknown if CMV status is just a biomarker of poor COVID outcomes or a driver/subsequence of SARS-CoV-2 induced disease. However, the increased odds ratio of CMV seropositivity is evident and may be useful in dictating potential prognosis of COVID patients.

The manuscript is written clearly and the conclusions are well supported by the data offered. I have no significant issues with the work as presented.

Reviewer #3: Weber et al prospectively looked at COVID19 patients in Germany and found a correlation between CMV seropositivity status and severity of COVID19. This is in addition to comorbidities and age. Surprisingly, they found that CMV seropositivity correlated with more severe outcomes in people younger than 70. Although this is an interesting finding, the numbers of subjects used in this study are relatively low. Would this finding hold true now in the omicron phase? Table 2 (comorbidities and age correlate with COVID19 severity) is already known so really it comes down to a single table/figure in this paper. Does correlation equal causation? They suggest the possibility of cross reactive T cells but their supplemental figure only has a few events that might show this. If so, what are they recognizing? This needs additional data/experiments.

6. PLOS authors have the option to publish the peer review history of their article (what does this mean?). If published, this will include your full peer review and any attached files.

Reviewer #1: No

Reviewer #2: No

Reviewer #3: No

---

## [Author Response · Author response to Decision Letter 0]

30 Apr 2022

Reviewer comments

Reviewer #1 (Remark to the authors)

In the here presented study by Weber et al., the authors investigate whether CMV serostatus can affect the outcome and severity of a subsequence SARS CoV-2 infection as a predictive correlation of CMV serostatus and hospitalization or severe outcome during Covid-19 disease could influence observation and treatment decisions in hospital setting worldwide. The authors do observe, that younger patients that go on to develop mild symptoms are almost exclusively CMV-seronegative whereas the CMV positive population seems to have a more severe outcome after SARS CoV-2 infection with increased numbers of hospitalizations and ICU visits. Due to increasing numbers of co-morbidities, this statistical significance does not seem to be present in the oldest age group in this study, indicating the CMV serostatus might be especially important for younger otherwise healthy patients, for which the largest effect of serostatus and disease outcome could be observed. The data presented in this study is highly interesting and unsurprisingly very relevant, but while the authors try to explain their sole focus on CMV with the unique immunology describe for this virus, which could affect the adaptive immunity in CMV positive individuals, I believe it would be important to include additional, hopefully non-significant, controls into the study as the claims made by the authors and the potential significance for the here presented data on the filed requires adequate data to strengthen the here postulated hypothesis.

We appreciate that the Reviewer acknowledges the scientific relevance of our work. We also thought about the analyses of additional chronic latent viruses commonly present among the human population to strengthen our findings on the role of pre-existing CMV infection in the course of COVID-19 disease. Because of the very high expected prevalence of most of the other herpesviruses (EBV, VZV and HV6, at least 90%), we analyzed only HSV, despite we expected that conclusions that can be drawn from such data are going to be limited. Nevertheless, these analyses were indeed ongoing and we now integrate them with the revision step. 

Major comments.

1) “To our surprise, in follow up experiments we could identify a strong and robust cytokine response to human Cytomegalovirus (CMV) pp65 peptide mix in different TCRs specific to SARS-CoV-2 S-protein derived from an ICU COVID-19 patient (Supplementary Fig.1 a-b).”

While I have no problem believing that this data is true, what makes the authors think that this is something specific to CMV? Could there be cross reactive T-cells to other herpesviruses? HSV-1 or EBV for instance? Could they have a similar effect? Honestly, I would likely advise the authors to also test all the sera shown in figures 1, 2 and 3 for IgG against other herpesviruses for the purpose of demining if they also might correlate with severe Covid-19 disease or not. At a minimum, this would serve as a control for their CMV studies showing that what they see is not universally true for all tested pathogens.

Like the Reviewer, we also considered the possible cross-reactivity to other herpesviruses. For this reason, we had evaluated the reactivity of primary T cells engineered with SARS-CoV-2-specific TCRs to an EBV peptide pool, and no responses were observed. This information has not been included in this manuscript, as it had been already shared with our previous work (Fisher et al., Nat Commun, 2021). However, we stressed out the observation in the revised version of the manuscript.

As suggested by the reviewer, we tested all sera for IgG against HSV (both 1 and 2), for which we expected a seroprevalence of around 70%. For other herpesviruses like EBV, VZV and HV6, we expected a seroprevalence above 90%, meaning that the low number of seronegative individuals would not allow drawing any conclusion. HSV seropositivity represented a risk factor for severe COVID-19 but particularly for middle-advanced aged patients. Notably, for young individuals with no comorbidities, CMV remained the only predictor of worse prognosis. The newly generated data for HSV were added to the revised version of the manuscript. 

2) “While primary and latent CMV infections in immunocompetent individuals do not cause major symptoms, CMV (re-)activation is a feared complication in immunocompromised patients and new-borns13–15”.

While CMV re-activation is clearly a major health concern in transplant recipients and can be problematic in congenital infections, primary infection of CMV naïve mothers resulting in congenital infection are probably the more impactful threat to neonates.

We thank the reviewer for raising this point. We included this in the main text of the revised version of the manuscript.

3) “Recently, a few cases of CMV reactivation in the setting of severe COVID-19 have been reported16–19.”

This statement is a little unclear. Have the CMV re-infections been observed as the results of SARS CoV-2 infections and the resulting Covid-19 disease, or could these reactivations have been the results of the treatment the patients received as a result of their condition which could have included steroids and hence might have cause an immunosuppressive environment in these individuals, which might have resulted in CMV re-activation as well as potentially the re-activations of other herpesviruses.

We agree with the Reviewer that this statement needs more clarification. On the one hand, it is true that severe COVID-19 patients receive steroids to suppress immunopathology, and this might induce an immunosuppressive state that could promote CMV-reactivation. On the other hand, severe SARS-CoV-2 infections induce lymphopenia and abundant release of proinflammatory cytokines known to be associated to CMV-reactivation. Thereby both options are hypothetically valid but none of them has been verified so far. The studies cited in the manuscript showed the first evidence of CMV re-activation in severe COVID-19. However, they included only a limited number of patients; therefore, conclusions on the reasons of the occurrence of CMV re-activation were not possible. This additional explanation has been added to the main text.

4) “Therefore, it was speculated that the development of effective T cell responses upon infection with SARS-CoV-2 could be strongly dampened by CMV-driven immunosenescence26,27, which might at least in part explain the high prevalence of severe disease in the elderly (>80 years).”

This is a very hypothetical statement, to my knowledge, CMV induced immunosenescence is a concept that has been postulated for humans, but has so far only been shown in inbred rodent model systems and is still a matter of some controversy. It is probably more likely that for the data presented here, co-morbidities and overall health might play a more significant role.

We agree with the Reviewer that CMV-driven immunosenescence is more postulated than demonstrated in humans. We made this clearer by rephrasing in the revised version of the manuscript.

5) “Overall, the identification of SARS-CoV-2/CMV cross-reactive T cells, the known impact of CMV infection on the immune system, as well as the first reports on CMV reactivation during severe COVID-19 guided us to investigate whether CMV seropositivity is associated with severe COVID-19.”

This is in my opinion very circumstantial evidence for an involvement of CMV in determining disease severity after SARS COV-2 infection. While the data generated in this study does look interesting, I would advise including other viruses (herpesviruses) as controls to clearly demonstrate the unique role and biology of CMV.

The only point we can currently make based on the provided data on SARS-CoV-2/CMV cross-reactivity is the fact that such cross-reactivity can exist (we are not aware that this has so far ever been shown so precisely by TCR engineering before). As this finding was the main reason for us to look more deeply into a correlation of CMV seroprevalence and COVID-19, we added it to the manuscript. We have initiated more extensive studies on cross-reactivity also including other viruses. Such experiments are technically highly challenging and some time will be required to provide a more generalizable picture on cross-reactivity. 

6) “Looking at CMV serostatus within different disease severities and decades of age further demonstrates that particularly younger patients who required admission to the ICU were mostly CMV seropositive, while this finding weakened with increasing age (Fig. 1).”

While younger patients are generally in better health than older individuals and hence have to be hospitalized less and spend less time in the ICU, does the here presented data indicate that younger individuals of lower socioeconomic status or from developing nations are at higher risk of more severe Covid-19 disease compared to their age matched peers in richer and more developed nations as they are more likely to be CMV seropositive? It’s here any data for this in the literature?

Reviewer´s speculation is logical. CMV seroprevalence has been associated to socioeconomic status, with low-income countries showing a higher incidence of CMV infections. In addition, individuals with low socioeconomic status seems to be more susceptible to severe SARS-CoV-2 infections, in particular in some racial/ethnic minority groups (Khanijahani et al. 2021; Magesh et al. 2021; Arceo-Gomez et al. 2021). Therefore, it is logical to suppose that the younger population in our cohort may have a lower socioeconomic status and, more broadly, that CMV serostatus may simply identify a minority with higher probability of developing severe COVID-19 due to its wellness background. However, we have no information helpful for the quantification of the socioeconomic status of our study participants; thereby no statements can be done in this regards. Still, this type of analyses are highly relevant, as discussed in the revised manuscript.

7) “Remarkably, all but one patient younger than 70 years admitted to the ICU and most hospitalized patients were CMV seropositive. Conversely, the CMV prevalence in the mild disease subgroup was similar to the age-matched healthy population in Germany28.”

While the authors show data indicating the CMV does have an effect on the severity of the disease, did it also affect the overall length of the stay in the hospital? Do the authors have any more hard virological or immunological data, e.g. viremia that would indicate that CMV status affects the subsequence Covid-19 disease progression?

We agree with the Reviewer on the relevance of analyzing CMV serostatus in relation to additional clinical parameters e.g. length of hospitalization and viremia. However, behind the fact that these analyses are not in the scope of this manuscript, the absence of young ICU patients with CMV negative serology prevents to draw any conclusion from these comparisons.

8) “Intriguingly, after age stratification, younger patients suffering from comorbidities (<70, Fig 3, node 3) were more likely to develop a severe course of disease requiring ICU treatment when CMV-seropositive (CMV positive: 33.3%; CMV negative: 4.0%) (Fig 3, nodes 7 and 8). In individuals without known co-morbidities, CMV seropositivity again served as a negative predictor of outcome, but was independent of age (node 5 and 6).”

Again, these data would indicate that younger people from lower socioeconomic backgrounds, especially in the poorest nations on earth additionally exposed to other circulating diseases like Mtb and other potential co-morbidities like malnutrition that could affect the overall health and immune status, should be more prone to higher hospitalization and death rates compared to their peers in the developed world. Is there any indication that this might be true?

We agree with the Reviewer on the importance of this argumentation, which has been already discussed in point 6.

9) “The identification of CMV/SARS-CoV-2 cross-reactive T cells (Suppl. Fig. 1) might indicate that CMV infection is indirectly involved in severe COVID-19 via the preferential recruitment of T cells from the antigen-experienced or memory T cell pool.”

While I do believe that that could be happening, the authors do not present any data that this is specific to CMV but simply work under that assumption. As mentioned above, some controls are advised.”

As already discussed in point 1, cross-reactivity to EBV was additionally checked without observing any T cell response upon EBV-derived peptide stimulation. Our observation at least support the evidence that cross-reactivity to CMV-derived epitopes can occur in SARS-CoV-2 infections, despite we cannot exclude that a similar effect might be triggered also by other herpersviruses not tested for T cell responses.

10) “Such T cells are often less reactive to the antigen for which they were not originally primed and because of this an impaired T cell response could fail to control SARS-CoV-2, thereby leading to severe COVID-19.?

It would be beneficial to the reader if the authors could give a reference for this statement.

We thank the Reviewer for this comment. A reference was implemented.

11) “Cross-reactivity to SARS-CoV-2 epitopes in severe COVID-19 patients has also been shown for other target specificities, such as other common cold corona viruses29–35.”

As mentioned before, this information should be grounds to test other herpesviruses especially EBV as an important control in this manuscript to determine if CMV is unique or not. When it comes to the T-cell responses, cross-reactive clones targeting other viruses apparently exist.

The reviewer´s comment was already discussed in points 1 and 9.

12) “Few recent case reports have described CMV-reactivation during SARS-CoV-2 and postulated that CMV-driven pneumonitis might have been a key driver of lung function compromise and clinical outcomes in these COVID-19 patients16,17,19.”

If this is true, would CMV be the result of the underlying SARS infection, or would CMV reactivation result from the steroid treatment during the severe course of infection and hence be independent of the ongoing virus infection?

As discussed in point 3, discriminating whether CMV-reactivation is due to COVID-19 or to the related treatments is crucial but it requires additional investigations and cannot be deducted from the existing data. 

Reviewer #2 (Remark to the authors)

The ongoing pandemic induced by infections with SARS-CoV-2 results in a wide array of disease outcomes ranging from asymptomatic to high morbidity and mortality. Multiple factors have been correlated with poor disease events including age, gender and pre-existing comorbidities. This current study seeks to characterize the impacts of seropositivity against CMV in relation to COVID disease. The authors are building on an earlier study to in which they enriched for T-cell receptors that recognize the coronavirus SPIKE protein from patients with severe COVID. The surprising finding was that there was a significant enrichment for TCR clones that and isolated clones had an increase reactivity profile for CMV antigens. This current study builds on this finding and seeks to determine the CMV seropositivity status of individuals in Germany that experience severe COVID outcomes in relation to those with milder symptoms. The authors provide compelling evidence that there is a significant correlation of CMV serostatus with poor COVID outcomes which was most evident in the younger population especially those with comorbidities. As CMV exhibits higher incidence with age, it was not surprising that a majority of the aged population with COVID were CMV seropositive. However when this group was stratified by CMV seropositive vs seronegative, there was not a significant increased risk factor for COVID hospitalization based on the presence of the herpesvirus.

It remains unknown if CMV status is just a biomarker of poor COVID outcomes or a driver/subsequence of SARS-CoV-2 induced disease. However, the increased odds ratio of CMV seropositivity is evident and may be useful in dictating potential prognosis of COVID patients.

The manuscript is written clearly and the conclusions are well supported by the data offered. I have no significant issues with the work as presented.

We are delighted about the reviewer’s positive evaluation of our work and the appreciation of scientific relevance of our study.

 

Reviewer #3

Specific points

Weber et al prospectively looked at COVID19 patients in Germany and found a correlation between CMV seropositivity status and severity of COVID19. This is in addition to comorbidities and age. Surprisingly, they found that CMV seropositivity correlated with more severe outcomes in people younger than 70. Although this is an interesting finding, the numbers of subjects used in this study are relatively low. Would this finding hold true now in the omicron phase? 

The Reviewer is right in asking whether our finding are true also for other SARS-CoV-2 variants, in particular the Omicron. Unfortunately, we only had access to biosamples collected in 2020 during the first waves of the pandemic where SARS-CoV-2 variants of concern had not established yet. This represents a limitation of our study cohort, which we discussed already in the original manuscript. Despite interesting, the suggested analysis is behind the scope of this manuscript. 

Table 2 (comorbidities and age correlate with COVID19 severity) is already known so really it comes down to a single table/figure in this paper. Does correlation equal causation? 

We disagree with the Reviewer. Table 2 contains both confirmatory and novel information, the latter in regards to the correlation between CMV status and COVID-19 severity. The univariate analyses showing odd ratios of known clinical parameters such as age and co-morbidities are known from the literature, but they represent important controls to validate the reliability of the cohort used in the study. On the contrary, analyses of CMV serostatus by univariate and multivariate models demonstrate for the first time that CMV seropositivity is an independent predictor of severe COVID-19. We then dived into this finding in the three main figures where we showed that, firstly, CMV serology is a strong predictor of worse COVID-19 prognosis in young patients and, secondly, how CMV serology may help in identifying individuals with high risk of developing severe COVID-19 in presence, and more importantly in absence, of other known co-morbidities (never showed before).

Despite the interesting new findings, our data cannot still dissect whether pre-existing CMV is a cause or simply a biomarker of a severe course of SARS-CoV-2 infection. Correlation is not equal to causation, but it represents the first indication that this could be a possibility and thereby can inspire future investigations. 

They suggest the possibility of cross reactive T cells but their supplemental figure only has a few events that might show this. If so, what are they recognizing? This needs additional data/experiments.

Similar to the reviewer, we also interrogated on which epitope of the CMV pp65 antigen the identified SARS-CoV-2-specific TCRs can cross-recognize. To do that, we used a matrix pools approach where overlapping peptides are generated from the target antigen and pooled into many subpools. Each peptide is present only in two subpools according to a designed layout. By applying this approach, we firstly identified the exact epitope cross-recognized by our TCRs and, secondly, we confirmed the specific recognition by peptide titration assays. The data were added to the revised version of the manuscript as new Supplementary Figure 1C-E.

---

## [Editor Report · Decision Letter 1]

3 May 2022

CMV seropositivity is a potential novel risk factor for severe COVID-19 in non-geriatric patients

PONE-D-22-02057R1

Dear Dr. Busch,

We’re pleased to inform you that your manuscript has been judged scientifically suitable for publication and will be formally accepted for publication once it meets all outstanding technical requirements.

Kind regards,

Juliet V Spencer, Ph.D.

Academic Editor

PLOS ONE
---

## [Editor Report · Acceptance letter]

17 May 2022

PONE-D-22-02057R1 

CMV seropositivity is a potential novel risk factor for severe COVID-19 in non-geriatric patients 

Dear Dr. Busch:

I'm pleased to inform you that your manuscript has been deemed suitable for publication in PLOS ONE. Congratulations! Your manuscript is now with our production department. 

Kind regards, 

on behalf of

Dr. Juliet V Spencer 

Academic Editor

PLOS ONE